# Inverse Problems Leveraging Pre-trained Contrastive Representations

**Sriram Ravula**∗
The University of Texas at Austin
Electrical and Computer Engineering
sriram.ravula@utexas.edu

**Georgios Smyrnis**∗
The University of Texas at Austin
Electrical and Computer Engineering
gsmyrnis@utexas.edu

**Matt Jordan**
The University of Texas at Austin
Computer Science
mjordan@cs.utexas.edu

**Alexandros G. Dimakis**
The University of Texas at Austin
Electrical and Computer Engineering
dimakis@austin.utexas.edu

## Abstract

We study a new family of inverse problems for recovering representations of corrupted data. We assume access to a pre-trained representation learning network R(x) that operates on clean images, like CLIP. The problem is to recover the representation of an image R(x), if we are only given a corrupted version A(x), for some known forward operator A. We propose a supervised inversion method that uses a contrastive objective to obtain excellent representations for highly corrupted images. Using a linear probe on our robust representations, we achieve a higher accuracy than end-to-end supervised baselines when classifying images with various types of distortions, including blurring, additive noise, and random pixel masking. We evaluate on a subset of ImageNet and observe that our method is robust to varying levels of distortion. Our method outperforms end-to-end baselines even with a fraction of the labeled data in a wide range of forward operators.

## 1 Introduction

Modern representation learning networks like CLIP [35] are showing incredible performance for image classification, even for zero-shot problems with labels not seen during training. Training these encoders comes at a staggering cost and requires datasets and computing resources only available to very few organizations. In this paper we show how to leverage this pretrained power for a new family of problems in the presence of image corruptions or other types of measurements.

Inverse problems involve reconstructing an unknown vector $x$ from measurements $y = A(x)$. Typically, the forward operator $A$ corrupts the unknown vector $x$ and reduces its dimension, i.e. the observations $y$ live in a lower-dimensional space compared to $x$. In the special case of linear inverse problems, the forward operator is simply a matrix and the measurements are in the form $y = Ax + $ noise. Special cases of linear inverse problems include image denoising, inpainting, super-resolution, compressed sensing used in medical tomography, seismic geological imaging and many others, see e.g. [31] for a recent overview.

---

∗Equal contribution.
Code available at https://github.com/Sriram-Ravula/Contrastive-Inversion.

35th Conference on Neural Information Processing Systems (NeurIPS 2021).

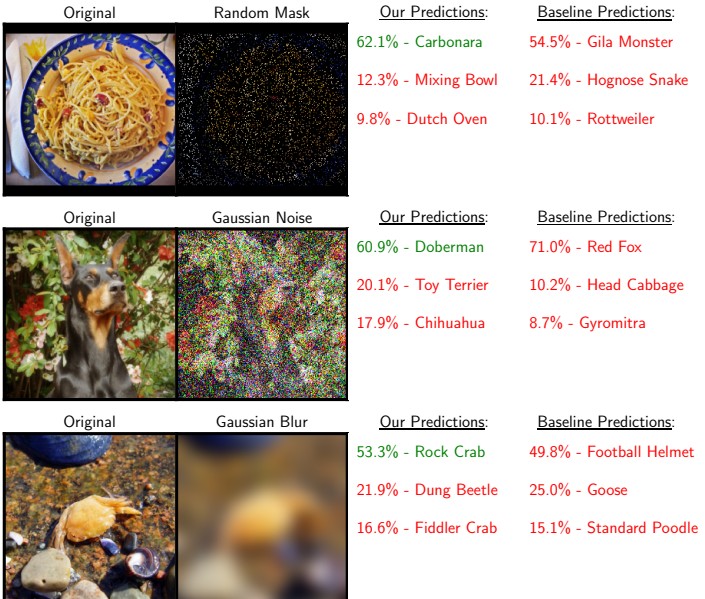

Figure 1: **Predictions of our models vs. supervised baselines for corrupted images.** Our robust encoders observe highly corrupted images and use a simple linear probe to classify in ImageNet-100 labels. We present the top 3 classes from our models as well as those from the end-to-end supervised baselines trained with the same amount of labeled data, for select images. For three different types of forward operators (90 percent missing pixels, strong Gaussian noise and high Blurring) our robust encoders classify correctly and also produce reasonable top 3 alternatives. On the contrary, the supervised baselines completely fail even though they were fine-tuned on exactly this task to classify corrupted images, starting from a powerful ImageNet pretrained ResNet-101. We also expect that most humans would fail to classify such highly corrupted images – more examples are included in the Appendix.

In this paper, we introduce the study of a new family of inverse problems: reconstructing the representation of an image given a corrupted or measured input. Formally, if a (clean) image is $x$ and its CLIP representation is $R(x)$, we would like to obtain that representation by only observing a highly corrupted input $A(x)$. This is impossible if the forward process $A$ removes information needed to obtain the representation. Surprisingly, we show that we can recover representations that are useful for downstream tasks, even from extremely corrupted versions of the image.

We introduce a robust encoder $S$ that is trained to imitate the behavior of the pretrained CLIP encoder acting on clean images $x$. However, the input to the robust encoder is only corrupted images $A(x)$ that are created by applying the forward operator on $x$. Our approach is illustrated in Figure 2. The teacher encoder is the pretrained CLIP, and the student encoder is our robust encoder operating on corrupted images. Formally, the robust encoder $S(A(x))$ is trained to approximate $R(x)$ using a contrastive loss.

Many applications, such as object recognition with low-cost cameras, remote sensing and aerial imaging rely on noisy or blurry data and can face occlusions or sensor corruptions. As we demonstrate in our experiments in the Appendix, normal CLIP fails on highly corrupted images. Our procedure allows us to transfer the power of CLIP to heavily corrupted images in downstream tasks, with relatively little extra training.

## 1.1 Results

We show that our method is able to obtain useful representations even under extreme corruptions such as removing $90\%$ of the pixels as shown in the top panel of Figure 1. The highly corrupted images enter our robust encoder and the obtained representation is used in a linear classifier to produce ImageNet-100 labels. Our main result is that our method outperforms a pretrained ResNet (of the same size as our robust encoder) fine-tuned end-to-end on labeled distorted images.

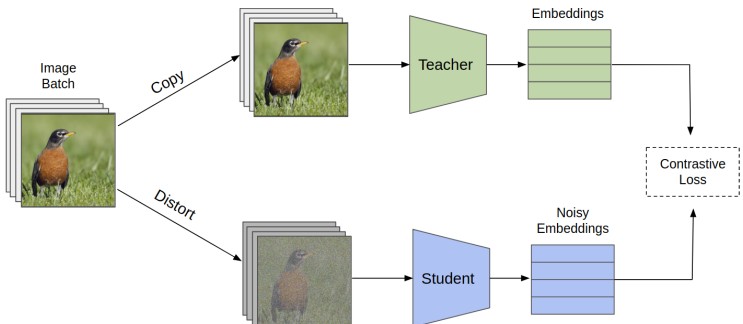

Figure 2: **Overview of our proposed method.** We initialize a student and teacher model from a pretrained CLIP encoder. Clean image batches are fed to the teacher while distorted versions of those images are fed to the student. The student is trained using a contrastive loss which makes student and teacher representations of the same original images more similar while making their representations of different images less similar.

**Using less labeled data** For some corruption levels, we are able to outperform end-to-end fine-tuned ResNets using as little as 10% of labeled samples. This is even when the fine-tuned baseline uses 100% of ImageNet-100 labels for training. The primary advantage of our model is that our robust encoder observes representations from the pretrained CLIP encoder, which was trained with a much bigger dataset compared to ImageNet. Still, the fact that this implicit advantage in the representations and 10% of labeled data is sufficient to outperform a supervised trained ResNet fine-tuned with 10 times more labeled data is very surprising and illustrates the power and versatility of pretrained representation learners.

**Robustness to noise and data shifts** Our method is very robust to changes in the forward operators, data statistics and label shifts. We experiment with three classes of forward operators: random pixel masking, additive Gaussian noise, and Gaussian blurring distortions. For each, we train and test on a wide-range of distortion levels, from slight to severe corruption. We show that our robust encoder produces useful representations even when the level of corruption is outside its training domain.

We show that our representations are useful for a wide range of tasks, without requiring knowledge of the task when the robust encoder is trained. We illustrate excellent classification accuracy across five datasets, frequently outperforming end-to-end supervised baselines trained with knowledge of the target task. Our experiments include a chest X-ray COVID pneumonia task which has very different morphology compared to ImageNet. Surprisingly, the same universal representations, combined with a custom linear probe are very successful across all tasks.

**Contrastive versus MSE training** We formulate the contrastive student training method as a regularization upon the simple mean squared error loss between student embeddings of a distorted image and teacher embeddings of the clean version of that image. We analyze the effects of this regularization on the training dynamics. Our results empirically show that simple MSE is worse in most cases, further strengthening our argument for the usefulness of contrastive learning in this setting.

## 2 Related Work

**Robust Image Recognition** It has been shown that classifiers that perform well on clean image tasks are not robust to common image distortions [21]. Several datasets have been proposed specifically to benchmark generalization of classifier performance to natural distortions [18, 38]. Related to our work, [45] and [12] fine-tune pretrained classifiers on distorted data, which seems to yield better performance than end-to-end training. However even under these training processes, modern classifiers exhibit inferior performance to human vision on distorted data [12].

**Inverse problems** There is significant recent literature on solving inverse problems including denoising, inpainting and deconvolution for deblurring. While classical techniques rely on sparsity based priors e.g. [30, 37], recent techniques include data-driven deep-learning methods [1, 10, 13, 33] as well as combinations of sparsity and generative methods [11] and untrained deep nets [16]. Our work focuses on recovering the representation of an image, as opposed to an image itself, so it is also related to task-aware sensing [23]. However, our approach is fundamentally different from all

previous inverse problems since instead of trying to reconstruct the image $x$, we aim to reconstruct a representation $R(x)$, which lives in a different space. The other important distinction is that even though our corruption processes are linear in the pixel space, they are non-linear with respect to the representation vector we try to recover. In effect, we are solving a non-linear inverse problem in a supervised way using a contrastive loss. In the Appendix, we compare to methods that attempt to solve the inverse problem in pixel space, then apply a classifier on the recovered image.

**Contrastive Representation Learning** Self-supervised representation learning has recently exploded in popularity, largely due to the success of constrastive losses in learning representations from unlabeled data [4, 6, 7, 15, 17, 43]. These techniques are able to generate highly general embeddings of images that are effective for many types of downstream tasks, even on domains that were not explicitly considered in training [27]. Contrastive losses generally operate based on a simple push-pull principle: images desired to be close in embedding space are pushed together, while unrelated images are pulled apart. One particularly popular choice of contrastive loss is the InfoNCE loss, derived from techniques for noise contrastive estimation [14], and popularized in the self-supervised setting in [32]. Several works have considered adversarial training frameworks to yield representations that are more robust to adversarial attacks [20, 22, 25]. However these works only consider adversarial robustness and not robustness to common corruptions. Our approach is similar to that of [24], which employs a variant of the InfoNCE loss for a supervised setting. This differs from our work in that we exclusively focus on robustness to natural corruptions. Moreover, our contrastive training step can be performed without any task-specific labeled data. This is made possible by the use of the powerful embeddings provided by CLIP, and allows the embeddings to be used for downstream tasks on multiple datasets.

**Knowledge Transfer Methods** Our work is closely related to prior works which aim to distill, reduce, or transfer the knowledge from one network to another for a specific task [3, 19, 29, 44]. Of note is [40], where the authors use a contrastive objective to transfer representations from a teacher network to a student network. Our work diverges in that we do not transfer from a larger, more powerful teacher to a smaller student, but rather transfer between a teacher and student of the same architecture initialized from the same weights. In addition, although the authors test on cross-modal transfer tasks such as transferring between color channels, transferring representations between clean and distorted images is a different task: we try to extract the same high-level information from *less data* as opposed to *different, but related data*.

## 3 Method

Our problem is to recover the embeddings of clean images when we only have access to highly corrupted versions of the images. The *encoder* $R(\cdot)$ is assumed to yield high-quality representations for a variety of domains. The *distortion process* $A(\cdot)$ is assumed to be a known forward operator that greatly distorts images. We assume $A(\cdot)$ is sufficiently severe as to inhibit the performance of the encoder $R(\cdot)$, but not so severe that recovery is impossible. From a collection of input images $\{x_i\}_{i=1}^N$, we are only given access to the distorted images and the representations of the clean images $\{A(x_i), R(x_i)\}_{i=1}^N$. Our approach is to learn a student function $S$ so that $S(A(\cdot))$ is equally useful as the teacher representation $R(\cdot)$. We measure the utility of a representation by the performance on an unspecified downstream supervised learning task.

**Least squares Loss** One potential approach for the task of recovering the teacher's embeddings from the corrupted inputs is to minimize the expected $\ell_2$ distance in embedding space between the clean teacher embeddings $R(\cdot)$ and the predicted student representations $S(A(\cdot))$. If both $R(\cdot)$ and $S(\cdot)$ are constrained to have $\ell_2$-normalized outputs, then the empirical least squares loss becomes:

$$\hat{\mathcal{L}}^{\mathsf{MSE}}(S; R, A) := \frac{-1}{N} \sum_{i=1}^N \langle S(A(x_i)), R(x_i) \rangle, \tag{1}$$

where we have expanded and subtracted out constant terms. However, this process may not yield the most effective embedding of the corrupted data. Due to either limitations in the training process, the severity of the distortion process, or the generalization properties of the approximate minimizers of $\hat{\mathcal{L}}^{\mathsf{MSE}}$, the learned embeddings $S(A(\cdot))$ may not be as useful for downstream tasks as embeddings learned using the other losses we consider.

**Contrastive Loss**   Inspired by recent advances in self-supervised representation learning, we learn $S$ by minimizing a contrastive loss. We consider the following variant on the popular InfoNCE loss:

$$\hat{\mathcal{L}}^{\text{contr}}(S;\tau,R,A) := \frac{-1}{N}\sum_{i=1}^{N}\log\frac{\exp(K(i,i)/\tau)}{\sum_{j=1}^{N}\exp(K(i,j)/\tau)}, \qquad (2)$$

where $K(i,j) := \langle S(A(x_i)), R(x_j)\rangle$ measures the similarity between the learned embedding of $A(x_i)$ and the clean embedding of $x_j$, and $\tau$ is a temperature hyperparameter. We follow [5, 42] and rewrite $\hat{\mathcal{L}}^{\text{contr}}$ in terms of explicit 'pull' and 'push' terms as :

$$\hat{\mathcal{L}}^{\text{contr}}(S;\tau,R,A) = \frac{1}{\tau}\hat{\mathcal{L}}^{\text{MSE}}(S;R,A) + \hat{\mathcal{L}}^{\text{unif}}(S;\tau,R,A), \qquad (3)$$

where the second term, referred to as the uniformity term, is defined as:

$$\hat{\mathcal{L}}^{\text{unif}}(S;\tau,R,A) := \frac{1}{N}\sum_{i=1}^{N}\log\sum_{j=1}^{N}e^{K(i,j)/\tau}. \qquad (4)$$

The first term of $\hat{\mathcal{L}}^{\text{contr}}$ is simply $\hat{\mathcal{L}}^{\text{MSE}}$ which encourages alignment of $S(A(x_i))$ with $R(x_i)$, and the $\hat{\mathcal{L}}^{\text{unif}}$ term encourages the learned representations for corrupted data to be dissimilar from all other representations of clean data.

The primary difference between Equation 3 and the InfoNCE loss commonly used in self-supervised learning is the choice of the similarity measure $K(\cdot,\cdot)$. Without access to specified target embeddings, $K(i,j)$ is typically chosen to be the inner product between projections of the embeddings of $x_i$ and $x_j$, (or between $x_i$ and a positive example in the case of $K(i,i)$). Here, the uniformity term is necessary to prevent representation collapse, where in our setting $\hat{\mathcal{L}}^{\text{MSE}}$ alone suffices to prevent this degenerate case. We note that alternative choices for $K(\cdot,\cdot)$ in $\hat{\mathcal{L}}^{\text{unif}}$ may be employed. For example, we could also contrast the student embeddings $S(A(\cdot))$ across two different images. We ablate against other choices in the experimental section and find that our choice of $K(\cdot,\cdot)$ is one of several that exhibits comparable performance.

**Effect of uniformity term**   To examine the effect of the uniformity term of the loss on the training dynamics, we consider the gradients of $\hat{\mathcal{L}}^{\text{contr}}$ with respect to the parameters of the encoder $S$. We first decompose $\hat{\mathcal{L}}^{\text{unif}}$ to consider the contributions of each individual data point:

$$\hat{\mathcal{L}}^{\text{unif}}(S;\tau,R,A) = \frac{1}{N}\sum_{i=1}^{N}\hat{\mathcal{L}}_i^{\text{unif}}(S;\tau,R,A), \quad \hat{\mathcal{L}}_i^{\text{unif}}(S;\tau,R,A) := \log\sum_{j=1}^{N}e^{\frac{K(i,j)}{\tau}}.$$

Then the gradient of $\hat{\mathcal{L}}^{\text{contr}}$ with respect to the parameters of $S$ may be written as

$$\nabla_S\hat{\mathcal{L}}^{\text{contr}}(S;\tau,R,A) = \frac{-1}{\tau}\nabla_S\hat{\mathcal{L}}^{\text{MSE}}(S;R,A) + \frac{1}{\tau N}\sum_{i=1}^{N}\sum_{j=1}^{N}w_i(j)\nabla_S K(i,j),$$

where

$$w_i(j) := \exp\left(K(i,j)/\tau - \hat{\mathcal{L}}_i^{\text{unif}}(S;T,R,A)\right).$$

This can be interpreted as follows. The weighting of the first term by $\tau^{-1}$ balances the gradients $\nabla_S K(i,i)$ and $\sum_j \nabla_S K(i,j)$, as is common with other choices of contrastive losses (c.f. Theorem 2 in [41]). Noting that for each $i$, the weights $w_i(j)$ sum to 1, the gradient of each individual uniformity term $\hat{\mathcal{L}}_i^{\text{unif}}$ is a convex combination of the gradients of the similarity terms $K(i,\cdot)$. The individual similarity terms are weighted exponentially proportionally to each $K(i,j)$'s contribution to the total uniformity loss. As $\tau$ decreases, the weights place greater emphasis on terms that are most similar. As $\tau$ is commonly chosen to be less than 1, the dynamics automatically reflect the influence of the 'hardest' negative examples.

**When is perfect recovery possible?** Finally, we describe conditions which allow for perfect recovery of the clean embeddings from the corrupted images in the training set. The first condition is that the corruption process should not be too destructive: it suffices to assume that the implication $A(x_i) = A(x_j) \implies R(x_i) = R(x_j)$ holds for all $x_i, x_j$ in the training set, i.e., there exists a function which attains exact recovery. In this case, any $S^*$ that minimizes $\hat{\mathcal{L}}^{\mathsf{MSE}}$ has $S^*(A(x_i)) = R(x_i)$ for all $x_i$ in the training set. To argue for the same recovery guarantees when optimizing $\hat{\mathcal{L}}^{\mathsf{contr}}$, we need to assume that the teacher $R(\cdot)$ provides a well-separated embedding of the training data.

**Proposition 1.** *If $R$ is a minimizer of the uniformity term*

$$R \in \arg\min_f \sum_{i=1}^{N} \log \sum_{j=1}^{N} \exp \frac{\langle f(x_i), f(x_j) \rangle}{\tau},$$

*then any encoder $S^* \in \arg\min_S \hat{\mathcal{L}}^{\mathsf{contr}}(S; \tau, R, A)$ exactly recovers the target embedding, $S^*(A(x_i)) = R(x_i)$ for all $x_i$ in the training set.*

**Training procedure** Our final method consists of (1) the contrastive step, in which the student learns representations for the distorted images and (2) a fine-tuning step, where we train a linear classifier on top of the learned representations. During the second step, the student encoder is kept frozen.

## 4 Experiments

We show that the proposed training process recovers useful representations from corrupted inputs for a variety of forward operators, evaluating several different classification tasks. We start by evaluating the representation quality when the distortion process and data distribution are the same during both training and inference, examining the label-efficiency of our approach.

Then we evaluate our approach when the severity of the distortions changes at test time, when the test data distribution differs from the one used during training, and when the labels are shifted. Finally, we ablate against alternative formulations of our loss function.

For all experiments, we perform contrastive training for the robust encoder using a 100-class subset of ImageNet, which we refer to as ImageNet-100, [36, 39] to reduce computational resources. Our target representations are attained from the CLIP ResNet-101. We find robust representations with contrastive learning and evaluate by training a linear classifier on top of the frozen representations. In all experiments, the distortions are applied randomly to each batch of images, and independently for each image in the batch. Where applicable, we report our results over 10 different instances of random corruptions applied on the evaluation images. Our baselines are built on a ResNet-101 initialized with weights from supervised training on the full ImageNet dataset. The final fully-connected layer is replaced with a 100-dimensional output. We fine-tune the whole model in a supervised fashion using distorted inputs and their correct labels from ImageNet-100. The baseline is trained for 25 epochs with a batch size of 64. Our robust encoder is trained for 25 epochs with a batch size of 256, and the linear probe on top of it is trained for 10 epochs. Additional experiments, as well as further details on training and hyperparameter choices are discussed in the appendix and in our provided code.

### 4.1 Known Data Distribution and Forward Operator

In this section, we evaluate the quality of the learned robust representations for classifying images from the validation set of ImageNet-100, using the same distortions during training and inference. These experiments demonstrate the usefulness of our method for vision inverse problems where the data distribution and forward operator are both known at training time. We also demonstrate the label-efficiency of our approach by evaluating our learned representation when only few labeled samples are available to train the linear classifier.

**Setup** We train the robust contrastive model and the baseline as described above. Eight different distortion processes are examined: Gaussian blur with (kernel size, standard deviation) of (21, 5) and (37, 9); additive Gaussian noise with standard deviations 0.1, 0.3, and 0.5; and random pixel mask with 50%, 75%, and 90% of the pixels missing. We evaluate our method against the baseline in top-1

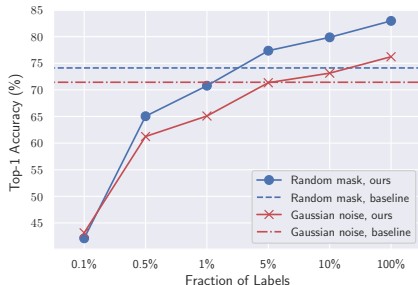

Figure 3: **Accuracies using a varying fraction of labeled samples to train a linear probe.** We train robust encoders on images with 90% random pixel masking and additive Gaussian noise with standard deviation 0.5, and fit a linear classifier on the learned representations using varying fractions of labeled training samples. We compare to a supervised baseline that uses all of the labeled training samples. Results are averaged over 10 random instantiations of corruptions on the ImageNet-100 validation dataset. We omit error bars as standard error is insignificant.

Table 1: **Top-1 accuracy (percent) on ImageNet-100**. The best accuracy for each distortion is bolded. Each model is trained using images with a fixed type of distortion. We train our robust CLIP encoder contrastively, then fit a linear probe on the learned representations using either all or 10% of the labeled training samples. We report the mean and standard error for accuracy over 10 random instantiations of distortions on the ImageNet-100 validation dataset (Gaussian blur is deterministic, so we do not include standard error values). For Gaussian blur, n corresponds to the length of the blur kernel.

| Distortion | Supervised Baseline | Ours | Ours (10% labeled data) |
|---|---|---|---|
| Random Mask 50% | 77.53±0.06 | **85.87±0.08** | 82.19±0.08 |
| Random Mask 75% | 75.68±0.06 | **83.99±0.07** | 80.36±0.09 |
| Random Mask 90% | 74.12±0.09 | **82.96±0.08** | 79.87±0.12 |
| Gaussian Noise $\sigma = 0.1$ | 82.23±0.04 | **84.46±0.08** | 80.99±0.09 |
| Gaussian Noise $\sigma = 0.3$ | 75.78±0.08 | **81.30±0.07** | 78.16±0.07 |
| Gaussian Noise $\sigma = 0.5$ | 71.43±0.14 | **76.23±0.10** | 73.14±0.08 |
| Gaussian Blur n = 21 | 76.40 | **83.24** | 80.94 |
| Gaussian Blur n = 37 | 68.94 | **77.80** | 74.84 |

accuracy on the validation set of ImageNet-100, under the same distortion used to train each model. To demonstrate the label-efficiency of our method, we also train a linear probe with only 10% of the labeled data, and for two of our models we train linear probes using various amounts of labeled data.

**Results**   In Table 1, we see that training a linear probe on top of the representations learned by our procedure greatly improves accuracy compared to the supervised baseline. This further solidifies our original motivation. Furthermore, we can see that using only 10% of labeled samples is sufficient for our model to outperform the baseline in most cases. More fine-grained label-efficiency results can be seen in Figure 3, where we show that in two of these cases, using just 5% of the labeled data makes our model outperform or be competitive with the baseline trained using all the labels.

### 4.2   Known Data Distribution and Unknown Forward Operator

In many settings, the data distribution and the *type* of distortion will be known at training time, but the *severity* of the distortion will be unknown. We consider the case where the severity of the distortion at test-time is greater than what was seen during training.

**Setup**   First, we train a model using training images with between 50 and 95% of pixels randomly masked. In addition, we train a model using images with additive Gaussian noise with random standard deviation between 0.1 and 0.3. Once fully trained, we fit a linear classifier on top of the learned representations for each model using the same distortions. We also train two supervised baselines end-to-end with the same distortions. We evaluate the models trained with pixel masking on images with a fixed level of 96 to 99% percent missing pixels, and the networks trained with noise on images with additive Gaussian noise using a fixed standard deviation between 0.35 and 0.5.

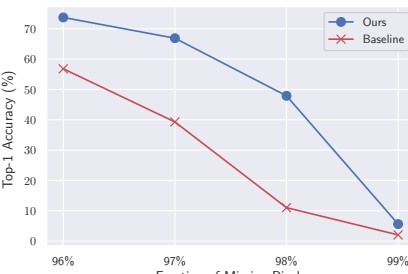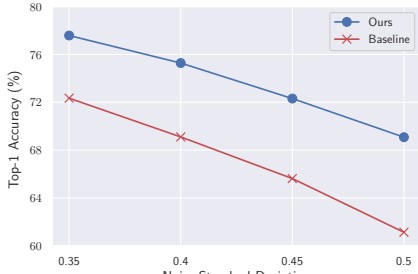

Figure 4: **Accuracies for images with varying corruption levels using models trained on a range of levels.** In the left figure, we compare our robust model with a baseline, both trained on images with 50% to 95% random pixel masking. In the right figure, each model is trained on images with additive Gaussian noise with random standard deviation from 0.1 to 0.3. We evaluate the models on images with more severe corruptions than applied during training. Results are averaged over 10 random instantiations of corruptions on the ImageNet-100 validation dataset. We omit error bars as standard error is insignificant.

Table 2: **Top-1 accuracies for transfer learning.** We fit a linear classifier for each dataset on top of the representations learned by the models from ImageNet-100. RM means the model was trained with random missing pixels, and GN means it was trained with additive Gaussian noise. Results are mean and standard errors over 10 realizations of the distortions during evaluation.

| Model | CIFAR-10 | CIFAR-100 | STL-10 | COVID X-ray | ImageNet-100B |
|---|---|---|---|---|---|
| Baseline (RM) | 79.83±0.04 | 55.10±0.08 | 81.09±0.05 | 82.07±0.23 | 67.45±0.09 |
| Ours (RM) | **80.93±0.04** | **58.55±0.09** | **89.44±0.05** | **84.01±0.20** | **80.09±0.07** |
| Baseline (GN) | **76.52±0.07** | 52.00±0.05 | 82.28±0.07 | **83.54±0.15** | 69.49±0.11 |
| Ours (GN) | 76.19±0.09 | **52.51±0.09** | **86.30±0.08** | 79.74±0.26 | **78.66±0.13** |

**Results** In Figure 4, we see that with an increase in noise levels, the accuracy of the models does decrease. However, the linear probe trained on the entire dataset achieves better results than the baseline end-to-end supervised model. This shows that our model is more robust, even when the distortions are greater than those expected during training.

### 4.3 Unknown Data Distribution and Known Forward Operator

In this section we evaluate how well ImageNet pretraining with distortions allows the learned representations to transfer to different datasets. We use the same forward operator during training and inference to isolate the quality of the embeddings learned during the contrastive step even in the presence of distortion.

**Setup** Five datasets are chosen to evaluate transferability of robust representations. (1) CIFAR-10 and (2) CIFAR-100 [28], and (3) STL-10 [8]. (4) The COVID-19 Chest X-ray dataset [9], (5) We generate another random 100-class subset of ImageNet [36] from the remaining 900 classes we did not use for ImageNet-100, which we refer to as ImageNet-100B.

We train the same models under the same distortions as outlined in Section 4.2, then fit linear classifiers for the new datasets on top of the fixed representations. Preprocessing details for each dataset may be found in the Appendix. We calculate top-1 accuracy of each model for classifying distorted images from the validation or test set of each of the new datasets. The images are distorted using the same forward operators used to train the networks.

**Results** Table 2 shows that our approach can achieve good results in a variety of datasets. In CIFAR-10 and CIFAR-100, we get results comparable to the baseline. In STL-10, we have greater top 1 accuracy for both noise settings. For the COVID X-ray dataset, we get mixed results, where we beat the baseline for the random masking model, but we lose for the additive Gaussian noise model. The most surprising result is the vast increase in accuracy for the alternative ImageNet-100B dataset.

Table 3: **Label Shift: Out of Distribution ImageNet Classes and their New Labels.** We list the five ImageNet classes taken from outside of ImageNet-100 and the new labels given to them based on similar classes from within ImageNet-100.

| ImageNet Class | New ImageNet-100 Label |
|---|---|
| Cup | Cocktail Shaker |
| Dungeness Crab | Rock Crab |
| Mountain Bike | Moped |
| Wood Rabbit | Hare |
| French Bulldog | American Staffordshire Terrier |

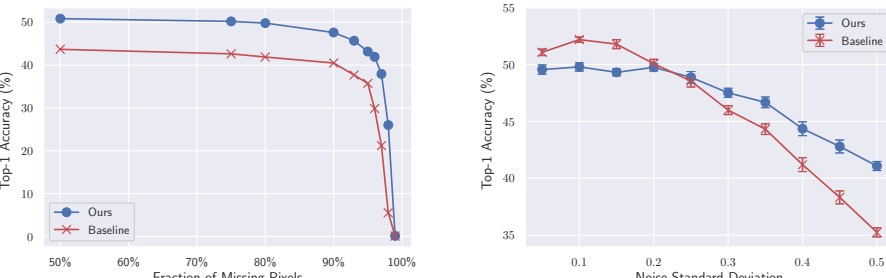

Figure 5: **Accuracies for varying noise levels on unseen classes using label shift.** On the left, we see the model trained on 50% to 95% random masking of pixels, while on the right the model trained on Gaussian noise with standard deviation from 0.1 to 0.3. Both models are evaluated on the unseen classes, using the chosen reference classes from ImageNet100 as the targets, as shown in the label shift table. Results are averaged over 10 random instantiations of corruptions. We omit error bars in the left figure as standard error is insignificant.

This shows that due to the supervised training of the baseline, some information from the labels leaks into the representation, leading to worse performance on a related, but ultimately different dataset.

## 4.4 Label Shift

An important factor in determining the robustness of a model is how gracefully it fails in the presence of unseen data modalities at inference time. For instance, if a network trained to distinguish dogs from cars is shown an image of a cat, the network should produce a embedding which is more likely to be classified as a dog than a car. This is even more important for inverse problems: a single distorted image may result from the same forward operator being applied to any number of original images. If a distorted image from a class outside of the training classes makes a network output an unexpected or uninformative representation, then the network is likely also brittle to shifts in the data distribution *within the classes it knows* at inference time. We evaluate the quality of representations produced by our method for distorted images from classes outside of the training dataset.

**Setup**   We use the same models described in Section 4.2. To simulate images from unseen classes, we identify 5 classes from the validation set of full ImageNet that were not used in our training data, but are similar to classes within our training data. We replace the labels of these new images with the label of the classes in our training data that are similar to them, as seen in Table 3. Evaluation is done on images from these replacement classes, under fixed levels of distortion, with 50% to 99% missing pixels on the random masking model and 0.05 to 0.5 standard deviation for the Gaussian noise model.

**Results**   In Figure 5, we can see that, for the random pixel mask case, we consistently outperform the baseline across all noise levels. For the Gaussian noise case, our model has slightly lower top-1 accuracy for small noises. This can be explained by the fact that small distortions do not drastically alter the image, so the baseline is bolstered by its ImageNet pretraining. However, as the noise level increases, the baseline model degrades much more quickly.

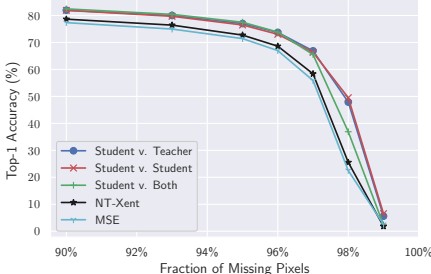

Figure 6: **Ablation study.** We evaluate the model trained on 50% to 95% random masking of pixels, using several variants of the uniformity term in the contrastive loss. Results are averaged over 10 random instantiations of corruptions on the ImageNet-100 validation dataset. We omit error bars as standard error is insignificant.

### 4.5 Ablation: Training Loss

Finally, we ablate against various choices for the uniformity term in the contrastive loss. We consider variants of $\hat{\mathcal{L}}^{\text{unif}}$ where we compare representations of i) the student and teacher, ii) the student and the student, iii) a sum of both of the above. We also consider $\hat{\mathcal{L}}^{\text{MSE}}$, i.e. no uniformity term, as well as one variant where every representation is contrasted with every other representation (irrespective of student or teacher), which we note is the same formulation as NT-Xent as used in SimCLR [6].

**Setup** We use the same random masking model described in Section 4.2, with a similar evaluation on fixed masking levels of 90-99%. Explicit formulations of each of the losses is in the appendix.

**Results** We draw two main conclusions from Figure 6. First, the MSE loss performs worst, indicating the benefit of the uniformity term in the loss. Even though we have access to the pre-trained representations, it is not simple for the model to exploit the encoded information. If this were the case, MSE would be as effective as our contrastive loss. Second, we see that all student comparisons perform roughly equivalently, but are clearly more effective than MSE and NT-Xent.

## 5 Conclusion

In this work, we propose a method for training image representation networks which are robust to various distortions on the input data. Our method has potential for improving the practical applications of powerful pre-trained models. Indeed, images in real-world settings are rarely exemplary: every stage of the imaging process, from capture to storage to transmission and display, can introduce noise or distortions in the images. Moreover, our process helps reduce the cost of training such models, both with respect to computation since we can add robustness to a pre-trained model instead of training one from scratch, as well as with respect to label efficiency since our method mainly relies on large amounts of inexpensive unlabeled data.

Several significant open problems emerge from this work. As we can see in Appendix D.8, our method requires some prior knowledge of the type of distortions that will occur to the image. This is due to the fact that our method relies on training the student to match the representations of the teacher, which is significantly more difficult when the type of corruption (and thus, the representation itself) changes drastically. As future work, we aim to extend our method beyond this limitation, possibly by finetuning the teacher as well as the student, to produce representations that are more easily matched under different types of distortions. In addition, another interesting research direction is how well this method performs under adversarially-chosen inputs.

## Acknowledgements

This research has been supported by NSF Grants CCF 1763702, 1934932, AF 1901281, 2008710, 2019844 the NSF IFML 2019844 award as well as research gifts by Western Digital, Interdigital, WNCG and MLL, computing resources from TACC and the Archie Straiton Fellowship.

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
