# OpenReview forum: "Inverse Problems Leveraging Pre-trained Contrastive Representations"
_NeurIPS.cc/2021/Conference — NeurIPS 2021 Poster_

### Official Review · Reviewer_oqRY · 2021-07-11

**Rating:** 6
**Confidence:** 4

**Summary:**

The paper consider the following problem: Let $x$ be an image, let $R$ be a feature map or representation obtained through the CLIP network, and let $A$ be a distortion process. The representation takes as input an image and yields a feature representation. The distortion process also takes an image as input, and yields a distorted image (e.g., it deletes pixels).

The paper assumes we are given the representations and corresponding distorted images, i.e., a collection $\{A(x_i), R(x_i)\}$, and the goal is to learn a student function $S(A( \cdot ))$ that is equally useful as the original representation $R(x)$ for a classification task. The paper learns the student $S$ by minimizing a contrastive loss, and the utility of the representation obtained by the student $S$ is measured by the performance on a supervised learning task, specifically on ImageNet-100.

The paper provides simulation results demonstrating that the method learns useful representations. The paper's main result is to show that the method 'outperforms a pretrained ResNet, of the same size as the robust encoder, fine-tuned end-to-end on labeled distorted images'. In addition to this result, the paper provides a number of ablations studies and compares to two reasonable baselines.

**Ethical Concerns:**

None.

**Limitations And Societal Impact:**

Yes.

**Main Review:**

Clarity: The paper is well written and well organized.

Quality: The paper provides good baselines and comparisons: The paper's main result is that the proposed method using the contrastive loss ''outperforms a pretrained ResNet, of the same size as the robust encoder, fine-tuned end-to-end on labeled distorted images''. Here, the paper compares to a ResNet that is trained on the noisy image end-to-end. That's a reasonable baseline; and another interesting baseline is to first denoise/recover the images and then train a classifier on the denoised images. This second baseline is studied in Appendix E.3.

Motivation/Significance: The motivation to study the problem is not entirely clear to me; the paper writes 'in this paper we introduce the study of a new family of inverse problems', but it is not clear to me where the problem considered arrises. Would it be possible to give a real-world example application where the problem considered occurs?

In summary, the paper introduces a new problem and provides a well-performing approach to address it, and also provides good baselines and comparisons. On the other hand, the motivation for the problem is entirely unclear to me which is a major negative point.


Minor comments:
- The paper provides as baseline a ResNet-101 which has the same number of parameters as the representation it learns, and the network is trained on the noisy examples. That's a good baseline; but would it be also possible to state the numbers for the same method (ResNet-101) trained on the original representation, to have an idea on how bad the effect of the respective perturbations is?
- The paper 'pretrains the robust encoder using the 100-class subset of ImageNet'. Would it be possible to add ablation studies showing how important the pretraining is for learning good representations?
- Regarding the setup; does the paper use the same noise and random pixel mask for each image, or a new one for each image? I assume a different one, but this wasn't clear to me from the main body, apologies if I missed something.
- The title 'inverse problems leveraging pre-trained contrastive representations' is misleading as it suggest the paper is concerned with solving an inverse problem. Yet, the paper is about learning a feature map from a disturbed version of an image.

**Time Spent Reviewing:**

4

---

> ### Author Response · Authors · 2021-08-10
> **Response to Official Review of Paper11153 by Reviewer oqRY**
>
> Thank you for your comments. We include our responses to the points you raised inline below.
> - **The motivation to study the problem is not entirely clear to me; the paper writes 'in this paper we introduce the study of a new family of inverse problems', but it is not clear to me where the problem considered arises. Would it be possible to give a real-world example application where the problem considered occurs?**
>   - Consider an underwater camera classifying different types of sea life, or a drone classifying aerial images. The images can be blurred or corrupted due to pollution, obstructions leading to missing data or sensor miscalibration. The labels also can be initially unknown or adjusted later and only a few labeled examples are available. We show that by contrastively training a network to reconstruct CLIP representations, we can fine-tune a model to classify highly corrupted images with very few labels (depending on the experiment, up to 10x compared to fully supervised methods). We thank you for your comment and we will include a detailed motivation in the revised paper (see also the related response to reviewer 3).
>
> - **Would it be also possible to state the numbers for the same method (ResNet-101) trained on the original representation, to have an idea on how bad the effect of the respective perturbations is?**
>   - Thank you for pointing out this comparison. We ran an experiment where we fine-tuned a linear probe using the distorted images, but on top of the original representation, and found out that it has worse performance. More specifically, we evaluated this technique when trained under fixed levels of noise, and evaluating on the same levels of noise. The results are as follows:
>
> |       Noise Type       | Acc +- Std. Error |
> |:----------------------:|:-----------------:|
> |     Random Mask 50%    |    46.3% ± 0.04   |
> |     Random Mask 75%    |    30.4% ± 0.04   |
> |     Random Mask 90%    |    23.1% ± 0.07   |
> | Gaussian Noise σ = 0.1 |    74.5% ± 0.05   |
> | Gaussian Noise σ = 0.3 |    50.8% ± 0.06   |
> | Gaussian Noise σ = 0.5 |    25.2% ± 0.09   |
> |  Gaussian Blur n = 21  |       65.8%       |
> |  Gaussian Blur n = 37  |       45.4%       |
>
> Compare the above to the results in Table 1 in the main paper. We see that these results are much worse than even the baseline, which shows that the perturbations have a significant negative effect on the quality of the representations used.
>
> - **Would it be possible to add ablation studies showing how important the pretraining is for learning good representations?**
>   - Please refer to our response to reviewer 2, where we pass corrupted images directly through CLIP with no contrastive training and fine-tune a linear classifier on the resultant representations. We can see that the contrastive step is crucial to learning good representations for the problem. This can also be intuitively gleaned from the supervised baselines, which lack this contrastive step.
>
> - **Does the paper use the same noise and random pixel mask for each image, or a new one for each image?**
>   - We use a different one for each image. We will make this clear in the paper.
>
> - **The title 'inverse problems leveraging pre-trained contrastive representations' is misleading as it suggests the paper is concerned with solving an inverse problem. Yet, the paper is about learning a feature map from a disturbed version of an image.**
>   - Indeed, we learn representations from perturbed images but writing it using a general forward operator is useful in our opinion: the formulation as an inverse problem makes the mathematical connections to other inverse problems (e.g. Compressed Sensing using Generative Models (Bora et. al, 2017), AmbientGAN (Bora et. al, 2018)) clear and can lead to the development of theory (e.g. sample complexity bounds) for general forward operators. The problem is different from other inverse problems since the reconstructed object is the representation $R(x)$, not the original image $x$ itself.

---

### Official Review · Reviewer_4CMh · 2021-07-15

**Rating:** 6
**Confidence:** 4

**Summary:**

This paper proposes an inversion method that exploits a pre-trained representation for regularization. Given only corrupted images, instead of working in the pixel space, it attempts to find their matching representation pairs in the pre-trained representation space that is trained with clean images by using contrastive objective function. Assuming that one has an access to a powerful representation that is already available, this method provides a robust representation against various types and levels of distortions and train/test time discrepancy.

**Ethical Concerns:**

No comment

**Limitations And Societal Impact:**

No comment

**Main Review:**

The organization of the paper needs to be revised. As a current form, it looks incomplete. The conclusion is missing, and the results are sparsely spread out over the paper. Considering the high academic standards, this is not acceptable, and I highly recommend the authors to revise the manuscript in a simpler and clearer manner.

- Other than the organization, my main concern is whether this is indeed a novel approach as the authors say. Please clarify the contribution and difference more against the works related to GAN inversion, deep generative compressed sensing (such as Bora et al. 2018), and [37].
  - Although it is not exactly the same, there have been several previous works in the field of GAN inversion such as deep generative prior that exploit the pre-trained GAN to find matching latent representations, which are eventually used for recovering the clean image given the (corrupted) measurements. On the other hand, AmbientGAN attempts to find a clean data distribution given only the corrupted measurement. In this regard, the proposed method is one branch of the existing problems sharing the motivation, and thus for me, the statement that this is a new inverse problem is an overclaim. However, I understand that their final goals are different in that the previous works aim for reconstructing an image and exploit the representation as a tool to regularize the solution space, while the proposed method aim for learning a new representation itself and use it for classification.
  - Line 105 – Line 107, “Our work diverges in that we do not transfer from a larger, more powerful teacher to a smaller student, but rather transfer between a teacher and student of the same architecture initialized from the same weights.” Isn’t this a too minor difference?
  - Line 30, “Surprisingly, we show that it can be very well approximated even from extremely corrupted versions of the image.” For me, this is also not surprising. Because a model has an access to the pre-trained representation of clean images of similar (sometime even the same) kinds, it can exploit the encoded information. It is indeed impressive that the representation is helpful even for classifying totally different data distribution such as medical data, this has been also well known as a typical advantage of having the pre-trained weights (a.k.a. warm starting).

- Line 44, “For some corruption levels, we are able to outperform end-to-end fine-tuned ResNets using as little as 10% of labeled samples. This is even when the fine-tuned baseline uses 100% of ImageNet-100 labels for training.”
Is it legitimate to say that the current method is using less supervision? CLIP is trained on massive amounts of data and these are all encoded in its representation, which means that the method is implicitly using/inheriting the encoded information. This needs some more discussions.

- Line 211, “We consider the case where the severity of the distortion at test-time is greater than what was seen during training.”
What happens when it is less severe at test-time?

- Only the classification is used as a downstream task to show the superiority of the learned representation. However, since the representation is learned against corrupted image measurements, it would be more interesting to see image reconstruction as a downstream task (how much it recovers the original image) using the learned representation.

- What is the payoff for having more robust representation against extreme corruptions such as removing 90% of pixels? In the related work, the authors said that adversarial training frameworks are robust to adversarial attacks but not to common corruptions. On the other way around, how’s this method against the adversarial robustness? Won’t this result in a bad adversarial robustness?


- Line 17, In this, paper -> In this paper,
- Line 98, Our approach is similar that -> similar to/with
- Line 177, Finally we -> Finally, we
- Line 178, Punctuation dot is missing at the end.

---

Update after rebuttal, the authors have addressed my concerns, and I increase my score to 6. However, as the other reviewers have pointed out, I also agree that the authors need to clarify the main contributions and differences of the proposed model against the related work with extra care. The organization of the manuscript also needs some changes.

**Time Spent Reviewing:**

8

---

> ### Author Response · Authors · 2021-08-10
> **Response to Official Review of Paper11153 by Reviewer 4CMh**
>
> Thank you for your comments. We include our responses to the points you made below.
> - **The organization of the paper needs to be revised. As a current form, it looks incomplete. The conclusion is missing, and the results are sparsely spread out over the paper.**
>   - Thank you for your feedback. We will revise the paper to include the conclusion and organize the results more compactly. We will also improve the motivation and clarify our contribution based on your and other reviewers’ feedback.
> - **Please clarify the contribution and difference more against the works related to GAN inversion, deep generative compressed sensing (such as Bora et al. 2018), and [37]. Although it is not exactly the same, ... while the proposed method aims for learning a new representation itself and use it for classification.**
>   - An inverse problem tries to reconstruct a vector from measurements $y = A(x)$, where $A$ is the forward operator introducing some type of corruption or projection. Compressed Sensing using Generative Models (Bora et al., 2017) reconstructs $x$ from $y = Ax + \mathrm{noise}$ when $x$ is in the range of a generative model. AmbientGAN (Bora et al., 2018) reconstructs the distribution of x (but not individual samples $x$) from corrupted measurements $y = A(x)$. The fundamental difference between our approach and these prior works leveraging deep networks for inverse problems is that the target we seek to recover is a representation of the original signal and not the signal itself (or a distribution of it). Because of the surjectivity of the encoder, we believe this to be an easier problem: we find that it is significantly more challenging to recover the true signal, and as such, the problem (and proposed solutions) are crucially quite different.
>   - While the approach in [37] appears similar to ours, we clarify the major differences. One key component of our approach is the ability to fine-tune a model, which is unavailable when compressing a model to a smaller architecture (see also our response to the below point). Perhaps our approach could be viewed as a form of cross-modal transfer, where we transfer between the domain of clean images to corrupted images, though we prefer the perspective of inverse problems. This way, we highlight connections with other forms of inverse problems, as well as potentially interesting future work. We also point out that the loss function that we consider differs from that of [37], specifically relating to the way we handle negative examples.
>
>     We will further clarify these points in the paper by including the above discussions.
>
> - **Line 105 – Line 107, “Our work diverges in that we do not transfer from a larger, more powerful teacher to a smaller student, but rather transfer between a teacher and student of the same architecture initialized from the same weights.” Isn’t this a too minor difference?**
>   - We do not feel that this is a minor difference from [37], since our robust student encoder will have the same capacity as the teacher model. This is key, since this means that our new model is able to theoretically minimize the contrastive loss and learn the exact teacher representations of the original, clean images (proposition 1). If we use a smaller student, then it may not have the capacity to minimize the contrastive loss.
> - **Line 30, “Surprisingly, we show that it can be very well approximated even from extremely corrupted versions of the image.” For me, this is also not surprising. Because a model has access to the pre-trained representation of clean images of similar (sometimes even the same) kinds, it can exploit the encoded information. It is indeed impressive that the representation is helpful even for classifying totally different data distributions such as medical data, this has been also well known as a typical advantage of having the pre-trained weights (a.k.a. warm starting).**
>   - We believe that the surprising part of our result is the usefulness of our trained robust encoder given the sheer scale of some of the corruption processes (see, e.g. Figure 5 in the appendix). The model is able to perform extremely well despite the presence of severe degradations that even humans may have a difficult time parsing. In addition, even though we have access to the pre-trained representations, it is not simple for the model to exploit the encoded information. If this were the case, simple MSE training loss would be as effective as our contrastive loss. Our contribution in this case is demonstrating a training procedure that is effective in taking advantage of the powerful pre-trained representation. We will clarify this in the main paper.
> - **Is it legitimate to say that the current method is using less supervision? CLIP is trained on massive amounts of data and these are all encoded in its representation, which means that the method is implicitly using/inheriting the encoded information. This needs some more discussions.**
>   - By less supervision, it is meant that we need far fewer labeled examples for the target task. Indeed, during the contrastive training phase, we require no task-specific labels, and we are more label-efficient when finetuning for downstream tasks. While it is true that we implicitly rely upon the massive corpus of CLIP's training data, we only require access to a strong encoder, which may be taken off-the-shelf without explicit access to the training data.
> - **“We consider the case where the severity of the distortion at test-time is greater than what was seen during training.” What happens when it is less severe at test-time?**
>   - We performed experiments where the noise levels were less severe than those seen during training and we see that our method still outperforms the baseline. More specifically, we evaluated the baseline and our method for this particular setting (corresponding to Figure 4) on noise levels lower than those used during training. The results are as follows, for the experiments with multiple levels of random masking and multiple levels of gaussian noise:
>
> | Random Mask  |   Baseline   |     Ours     |
> |:------------:|:------------:|:------------:|
> |      30%     | 76.75 ± 0.05 | 86.06 ± 0.06 |
> |      35%     | 76.89 ± 0.07 | 86.04 ± 0.05 |
> |      40%     | 76.93 ± 0.04 | 86.11 ± 0.06 |
> |      45%     | 77.05 ± 0.08 | 86.06 ± 0.05 |
>
> | Gaussian noise σ  |   Baseline   |     Ours     |
> |:-----------------:|:------------:|:------------:|
> |        0.02       | 77.92 ± 0.02 | 85.71 ± 0.02 |
> |        0.04       | 80.04 ± 0.05 | 86.13 ± 0.04 |
> |        0.06       | 80.95 ± 0.05 | 86.40 ± 0.04 |
> |        0.08       | 80.98 ± 0.04 | 86.25 ± 0.07 |
>
> These results are in line with the rest of the observations in this setting - accuracy for both models is higher (due to lower noise), and our method still outperforms the baseline.
>
> - **Since the representation is learned against corrupted image measurements, it would be more interesting to see image reconstruction as a downstream task (how much it recovers the original image) using the learned representation.**
>   - A setup for this problem may look like the following, if we understand correctly: using a powerful pre-trained VAE, learn a robust student encoder to recover the latent representation of corrupted images from the pre-trained teacher encoder. Then recover the images by passing corrupted images through the student, then passing the resultant latent representation through the (fixed) decoder, and then evaluate the quality of the recovered images. We present a classification version of this type of experiment in the appendix in Table 6, where on top of our contrastively learned representations we use a linear classifier trained on clean images, instead of a decoder. Our main goal with this work is not to try to characterize the original image distribution - for tasks like this, GAN inversion methods may work better.
>
> - **What is the payoff for having more robust representation against extreme corruptions such as removing 90% of pixels? In the related work, the authors said that adversarial training frameworks are robust to adversarial attacks but not to common corruptions. On the other way around, how’s this method against the adversarial robustness? Won’t this result in a bad adversarial robustness?**
>   - There are numerous uses for robust classifiers. Consider an underwater camera classifying different types of sea life, or a drone classifying aerial images. The images can be blurred or corrupted due to pollution, obstructions leading to missing data or sensor miscalibration. The labels also can be initially unknown or adjusted later and only a few labeled examples are available. We show that by contrastively training a network to reconstruct CLIP representations, we can fine-tune a model to classify highly corrupted images, with the benefit of requiring fewer labeled samples for the downstream task (depending on the experiment, up to 10x compared to fully supervised methods). We thank you for your comment and will include a more detailed motivation in the main paper (see also answer to reviewer 4).
>   - For the second question, we have not tested the robustness of our method to adversarial attacks. It is possible that CLIP representations are removing non-robust features (and hence our method is more robust) but any such claim would require significant further study. We will briefly mention this as an open problem based on your suggestion.

---

### Official Review · Reviewer_N8iG · 2021-07-17

**Rating:** 7
**Confidence:** 3

**Summary:**

This works suggests a novel method on how to harness the representational power of models such as CLIP. In this task the input is distorted with a known function, such as missing pixels or gaussian noise, blurring. This works proposes training a student contrastively by matching the clip representation of the clean image with student's representation of the distorted image. They propose different matching functions and explain why in their setup it will not collapse if one does not compare against every other.
The experimental setup is extensive and shows much better label efficiency when working with distorted images, they also show better noise extrapolation and better dataset tranferability in compare to baseline. Their baseline is a resnet pretrained on imagenet supervised trained on the distorted imagenet-100.

**Limitations And Societal Impact:**

Yes, they have discussed the main limitations in the appendix. This is a strong paper and I will increase the score with the CLIP on distorted results. Most probably this work outperforms just running CLIP on the distorted images directly, but currently it is an importantly missed baseline.

**Main Review:**

The main idea here, contrastively matching clean clip representations, is novel for this specific task. similar ideas have been used for adversarial robustness as well as related work explained in the paper.

One question is the choice of CLIP as the teacher. It would be interesting to know how much of the gain is to just from matching "A" clean representation and how much of it is from "The best" clean representation. An experiment with for example a simclrV2 pretrained as the teacher can give some insight whether just having access to the clean images during training gives the same gain.

They also propose a specific similarity measure and explain how it prevents collapse and tends toward hardest negatives. Their ablation also suggests that their proposed formulation is slightly better than simple MSE or typical InfoNCE loss (simCLR style).

The main portion of the paper is dedicated to experiments. They have extensive evaluation which shows their method consistently outperforms their baseline. The only baseline that they consider is a resnet pretrained on imagenet, finetuned supervised on the distorted dataset. Another important baseline which is missing is passing the distorted images through clip directly and finetuning a linear prob.

Apart from the label efficiency experiment results which are super impressive, I was also impressed by their results on OOD class accuracy and robustness.

In general, this is a significant step in utilizing the representation power of giantly trained models such as clip for computer vision tasks. The writeup is dense specially in the method section, but the experimental setup and results are properly presented and discussed. Unfortunately, the limitations (conclusion) section is forced to appendix.

----Post rebuttal

Thank you for running the experiment with CLIP on corrupted images. The results further strengthens the claims in this paper, therefore I'm increasing my score.

**Time Spent Reviewing:**

4

---

> ### Author Response · Authors · 2021-08-10
> **Response to Official Review of Paper11153 by Reviewer N8iG**
>
> We thank you for your comments and positive feedback on our paper. A few points we want to make can be found below.
>
> - **One question is the choice of CLIP as the teacher. It would be interesting to know how much of the gain is to just from matching "A" clean representation and how much of it is from "The best" clean representation. An experiment with for example a simclrV2 pretrained as the teacher can give some insight whether just having access to the clean images during training gives the same gain.**
>   - Our reasoning for using CLIP as a teacher is that it is a powerful pre-trained network that works with remarkably general labels -- we did not want to simply measure performance on ImageNet. However, the effect of having the same images at both pre-training and robust contrastive training time is an interesting direction we will discuss.
> - **Another important baseline which is missing is passing the distorted images through clip directly and finetuning a linear probe.**
>   - Thank you for this suggestion. We ran experiments passing corrupted images through a fixed CLIP Resnet-101 backbone and fine-tuning a linear classifier on these representations. We train and evaluate on ImageNet-100. The results are presented in the following table:
>
> |       Noise Type       | Acc. +- Std. Error |
> |:----------------------:|:------------------:|
> |     Random Mask 50%    |    41.3% ± 0.06    |
> |     Random Mask 75%    |    24.0% ± 0.07    |
> |     Random Mask 90%    |    14.5% ± 0.10    |
> | Gaussian Noise σ = 0.1 |    75.2% ± 0.06    |
> | Gaussian Noise σ = 0.3 |    25.1% ± 0.07    |
> | Gaussian Noise σ = 0.5 |     7.7% ± 0.09    |
> |  Gaussian Blur n = 21  |       51.2%        |
> |  Gaussian Blur n = 37  |        22.3%       |
>
> Compare these results to the bolded column in Table 1 in the main paper. As expected, this training setup performs very poorly compared to our method and even the fully-supervised baseline. This indicates that the base CLIP model does not create good representations for corrupted images, and that further training is needed to make the encoder more robust.

---

### Official Review · Reviewer_9uEJ · 2021-07-23

**Rating:** 5
**Confidence:** 3

**Summary:**

This paper proposes to recover the feature representation, pre-trained from CLIP, of a clean input image given a corrupted version of the image. They propose to recover the features, instead of the clean image, as their goal is to use these features for downstream tasks, e.g. classification. Furthermore, this allows for the transfer of knowledge from CLIP, which reduces the amount of labeled data needed for the downstream task. To recover the features, they train a student model via L2-Loss and contrastive loss given the clean feature from CLIP. Empirically, they evaluate on a subset of ImageNet (with only 100 classes) and consider three types of corruptions, random masking, Gaussian noise, and Gaussian blur. Empirical comparison with a baseline method, trained end-to-end with corrupted images, demonstrates their approach is more robust and requires less labeled data.

**Limitations And Societal Impact:**

The authors adequately addressed the limitations and potential negative impact.

**Main Review:**

# A. Originality

The task presented in this work is interesting, i.e., the reconstruction of the cleaned pre-trained features from CLIP. Prior works in inverse problems tend to focus on recovering ``good visual quality’’, however, when applying to downstream tasks, direct recovery of features seems to be a good strategy. However, the proposed approach’s method is based on existing work on contrastive loss, and fairly similar to Supervised contrastive learning (NeurIPS 2020) [23]. While the authors explained

> propose a fundamentally unsupervised approach where we have no knowledge of labels when learning the representations.

I find the difference minor, specifically, instead of label supervision, their approach requires pre-trained features from CLIP. Additionally, labels are required for the explored down-stream tasks.


# B. Quality
The writing of the paper is adequate, and the approach section is easy to follow. Below are some questions and concerns I have with the experiment setup and compared baseline.


## B1. Experiment Setup

* B1A) ImageNet Subset
The experiments are conducted using a subset of ImageNet using 100 classes. The authors cited [36] which uses the ImageNet100 dataset. However, keep in mind that in [36] they did report the full 1000-class ImageNet result. Reporting the full ImageNet result is beneficial as it allows for a fair comparison.

* B1B) ImageNet-C Benchmark.
In the main paper, no result for ImageNet-C (Hendrycks and Dietterich 2019) is presented. In the supplementary materials, results with a subset of Image-NetC is presented; however, once again, the subset is not ideal. The whole point of ImageNet-C is to

> establish rigorous benchmarks for image classifier robustness.

This will allow a fair comparison with prior works and more thoroughly evaluate the proposed approach.

* B1C) The reported error bar is on the randomness of the corruption at test-time.
The paper could be strengthened by also reporting the error-bar over the randomness during training, i.e., train a model multiple times with different random seed.


## B2. Baseline Model
* B2A) More details on how pre-training is done?
At Line 179, it states
> we pretrain the robust encoder using a 100-class subset of ImageNet
I couldn’t find more details on how the pre-training is implemented. Supplemental seems to describe the training details.

* B2B) The baseline model not pre-trained on ImageNet-100
At Line 182, it states
> Our baselines are built on a ResNet-101 183 initialized with weights from supervised training on the full ImageNet dataset

It would be better to also pre-train the baseline models on ImageNet-100, as it is the dataset of evaluation. Also, this would make the baseline method consistent with the proposed method. Otherwise, the effect of pre-training is not ablated from the proposed approach.

* B2C) Hyperparameter tuning on the reported dataset?
From the text, it seems like hyperparameters are searched on the same set as the reported result. Is there a separate validation set for tuning the hyperparameters?

* B2D) Please report the full hyperparameter search space
In supplemental Line 122

> We choose the hyperparameters for each model using a linear search over several values for each hyperparameter.

For completeness, please report all the searched values.


# C. Clarity
## C1. Title
I find the title not suitable for this paper. Specifically, the paper is more about building a robust classifier, as all the downstream tasks are related to classification and less about "inverse problems". Furthermore, the result on ImageNet100-C suggests that the approach isn’t beneficial in a few cases of corruption, hence the general catch all name of "inverse problems" seems to be an overclaim.

## C2. Motivation
The introduction of the paper could be improved. I recommend motivating why one would want to recover the clean pre-trained features; Currently, the motivation comes towards the end of Sec. 1.


# D. Significance
Overall, the paper demonstrates some potential of the proposed approach. However, there remain some concerns regarding the experiment setup and compared baseline as stated above.


**Time Spent Reviewing:**

6.5

---

> ### Author Response · Authors · 2021-08-10
> **Response to Official Review of Paper11153 by Reviewer 9uEJ**
>
> Thank you for your numerous suggestions to improve our paper. We include the comments and our replies inline.
> - **The proposed approach’s method is based on existing work on contrastive loss, and fairly similar to Supervised contrastive learning (NeurIPS 2020) [23]. While the authors explained “propose a fundamentally unsupervised approach where we have no knowledge of labels when learning the representations”, I find the difference minor, specifically, instead of label supervision, their approach requires pre-trained features from CLIP. Additionally, labels are required for the explored down-stream tasks.**
>   - The main difference to Supervised Contrastive learning [23] is that we only require representations, and not labels, during the contrastive training step. Since CLIP representations can be used for a very wide range of tasks, this is quite universal and only minimal fine-tuning is needed when the required labels are decided.  In contrast, [23] requires access to labels from the beginning of the process. Acquiring this many labeled samples may be expensive, and immediately invalidates unlabeled samples from being used during the contrastive procedure. Labels for the downstream tasks can be significantly fewer than the ones required for the contrastive step.
> This is possible because we benefit from CLIP as a powerful near-universal representation. We believe that the ability to leverage the power of these representations to classify corrupted images with fewer labels can be useful for numerous applications. We will emphasize this point in the paper.
> - **In [36] they did report the full 1000-class ImageNet result. Reporting the full ImageNet result is beneficial as it allows for a fair comparison.**
>   - We used ImageNet-100 primarily due to computational and energy constraints, as the subset of ImageNet considered is a sufficiently large and challenging dataset to allow comparisons to prior works. While the other 900 classes might cause a shift in top-1 accuracy for all models (as evidenced in the differences between Tables 1 and 9 in [36]), we do not believe this will affect relative performance across methods. It would be an interesting future ablation study to evaluate performance on datasets as the number of classes increases.
> - **In the supplementary materials, results with a subset of Image-NetC is presented; however, once again, the subset is not ideal.**
>   - As in the above point, we opted to use a part of ImageNet for computational, energy and cost efficiency. This does indeed slightly hinder the ability to make comparisons via ImageNet-C, but we believe that the experiments included in the main paper demonstrate the capabilities of our method.
> - **“we pretrain the robust encoder using a 100-class subset of ImageNet” I couldn’t find more details on how the pre-training is implemented. Supplemental seems to describe the training details.**
>   - Pre-training in this context refers to the contrastive training procedure we propose. The details for this step are described in lines 87-95 and 108-113 of the supplemental material, for the image preprocessing and hyperparameter selection respectively. To summarize, we apply a set of standard augmentations to the training images, as well as a set of standard optimizers and hyperparameter training (see below for more details for the latter). We will clarify this in the paper.
> - **The baseline model not pre-trained on ImageNet-100 … It would be better to also pre-train the baseline models on ImageNet-100, as it is the dataset of evaluation.**
>   - Thank you for the suggestion. We performed this additional experiment and indeed pre-training with ImageNet-100 makes the baseline worse. Specifically, instead of starting with a ResNet101 initialized with weights from supervised training on full ImageNet, we evaluate the baseline with one initialized with weights from supervised training from scratch on ImageNet-100. This supervised training was done for 90 epochs, with an SGD optimizer, learning rate of 0.1, momentum of 0.9 and batch size of 256 (the default hyperparameters for full ImageNet training as in the official pytorch examples repository, found here: https://github.com/pytorch/examples/tree/master/imagenet).  The results for the experiment with fixed noise levels (corresponding to Table 1) are as follows:
>
> | Noise Type | Extra Baseline | Baseline in paper |
> |:----------------------:|:--------------:|:-----------------:|
> |     Random Mask 50%    |  76.48 ± 0.02  |    77.53 ± 0.06   |
> |     Random Mask 75%    |  74.38 ± 0.07  |    75.68 ± 0.06   |
> |     Random Mask 90%    |  70.77 ± 0.10  |    74.12 ± 0.09   |
> | Gaussian Noise σ = 0.1 |  78.95 ± 0.06  |    82.23 ± 0.04   |
> | Gaussian Noise σ = 0.3 |  74.03 ± 0.08  |    75.78 ± 0.08   |
> | Gaussian Noise σ = 0.5 |  69.34 ± 0.09  |    71.43 ± 0.14   |
> |  Gaussian Blur n = 21  |      72.70     |       76.40       |
> |  Gaussian Blur n = 37  |      67.26     |       68.94       |
>
> For the experiment with varying noise levels (corresponding to Figure 4), the results are as follows:
>
> | Random Mask  | Extra Baseline | Baseline in paper |
> |:------------:|:--------------:|:-----------------:|
> |      96%     |  57.21 ± 0.11  |    56.80 ± 0.01   |
> |      97%     |  41.58 ± 0.07  |    39.28 ± 0.15   |
> |      98%     |  10.84 ± 0.09  |    11.01 ± 0.12   |
> |      99%     |   1.12 ± 0.02  |    2.03 ± 0.12    |
>
>
> | Gaussian Noise σ | Extra Baseline | Baseline in paper |
> |:----------------:|:--------------:|:-----------------:|
> |       0.35       |  71.71 ± 0.17  |    72.34 ± 0.12   |
> |        0.4       |  68.40 ± 0.09  |    69.10 ± 0.12   |
> |       0.45       |  62.75 ± 0.13  |    65.62 ± 0.12   |
> |        0.5       |  54.16 ± 0.10  |    61.13 ± 0.19   |
>
> These results are comparable to (and in most cases worse than) our original baseline. This is to be expected, since the model which is initialized with full ImageNet weights was trained on roughly 10 times more data than this new model. In any case, results are still worse than our method. This means that our method is capable of outperforming the stronger of the two baselines.
>
> - **B1C) The reported error bar is on the randomness of the corruption at test-time. The paper could be strengthened by also reporting the error-bar over the randomness during training, i.e., train a model multiple times with different random seed.**
>   - That is a good point. As mentioned previously, due to computational limits, we were unfortunately unable to include multiple trainings of the same model. We will include a clarification of this point in the paper.
> - **B2C) Hyperparameter tuning on the reported dataset? From the text, it seems like hyperparameters are searched on the same set as the reported result. Is there a separate validation set for tuning the hyperparameters?**
>   - We did not use a separate validation set to tune hyperparameters. Instead, we ran our model for a single epoch, and extrapolated the results in our final set based on the results of a single epoch. We will include a detailed explanation of this in the supplementary material.
> - **Please report the full hyperparameter search space.**
>   - Hyperparameter search was done over the following ranges:
>     - The learning rate for our method was searched in the range of [1e-4, 1e-2].
>     - For the baseline, the search was over [1e-5, 5e-1].
>     - The weight decay was searched in the range [1e-4, 1e-3].
>     - The temperature parameter τ was searched in the range [0.1, 1].
>     - The batch size was set as high as possible with our computing hardware. For our method, it is known that this form of contrastive loss benefits from greater batch size. For the baseline, this choice improved performance.
>
>   All this information will be included in the final paper.
>
> - **The paper is more about building a robust classifier, as all the downstream tasks are related to classification and less about "inverse problems". Furthermore, the result on ImageNet100-C suggests that the approach isn’t beneficial in a few cases of corruption, hence the general catch all name of "inverse problems" seems to be an overclaim.**
>   - We formulate our setup as an inverse problem since we want to reconstruct the CLIP representation from noisy or corrupted representations. We think that this view is useful since the connections to Compressed Sensing using Generative Models (Bora et al., 2017) and AmbientGAN (Bora et al., 2018) become clear and the process of corruption is general and described by the forward operator. This problem is fundamentally different from other inverse problems because the object recovered is the representation $R(x)$ as opposed to $x$ itself. Our formulation as an inverse problem is general and can benefit from the theory (e.g. provable sample complexity bounds) developed for inverse problems under generative models. The central benefit of the inverse problem formulation is the separation of the corruption process from the classification task. Practically, we enable classification from highly corrupted images, for general corruptions (forward operators) without needing to know the labels at the contrastive training step.
>
> - **The introduction of the paper could be improved. I recommend motivating why one would want to recover the clean pre-trained features; Currently, the motivation comes towards the end of Sec. 1.**
>   - Thank you for the suggestion. We will motivate the need for robust classifiers and the benefits of using fewer labels by leveraging CLIP earlier in the introduction.

---

### Author Response · Authors · 2021-08-10
**General Response to Reviewers**

We want to thank the reviewers for their insightful comments. Apart from answers to individual questions, we wanted to clarify some details about the terminology we use. More specifically, the training procedure of our method consists of the following two steps:
- A contrastive step, where we pass the representations from our teacher model to our student model.
- A fine-tuning step, where we train a linear classifier, while keeping the robust representations frozen.

For the teacher models, as well as the initialization of our baselines and student models, we download and use readily available, pre-trained models. The use of the word “pretrain” in lines 179 and 227 is an error, as in those lines we refer to the contrastive training step and not to the publically available model weights from OpenAI and PyTorch. We will clarify the differences between each stage of the training procedure more thoroughly in the main text.

---

### Decision · Program_Chairs · 2021-09-27

**Decision:**

Accept (Poster)

**Comment:**

While the reviewers criticize the quality of writing and the unusual terminology, the artificial nature of the proposed problem, they seem to be be in agreement about the novelty and significance of the results.